# Cost-effectiveness of a combined classroom curriculum and parental intervention: economic evaluation of data from the Steps Towards Alcohol Misuse Prevention Programme cluster randomised controlled trial

Ashley Agus,[1] Michael McKay,[2] Jonathan Cole,[2] Paul Doherty,[1] David Foxcroft,[3] Séamus Harvey,[4] Lynn Murphy,[1] Andrew Percy,[5] Harry Sumnall[6]

For numbered affiliations see end of article.

**Correspondence to**
Dr Ashley Agus;
ashley.agus@nictu.hscni.net

## ABSTRACT

**Objectives** This study aimed to assess the cost-effectiveness of the Steps Towards Alcohol Misuse Prevention Programme (STAMPP) compared with education as normal (EAN) in reducing self-reported heavy episodic drinking (HED) in adolescents.

**Design** This is a cost-effectiveness analysis from a public sector perspective conducted as part of a cluster randomised trial.

**Setting** This study was conducted in 105 high schools in Northern Ireland and in Scotland.

**Participants** Students in school year 8/S1 (aged 11–12) at baseline were included in the study.

**Interventions** This is a classroom-based alcohol education curricula, combined with a brief alcohol intervention for parents/carers.

**Outcome measures** The outcome of this study is the cost per young person experiencing HED avoided due to STAMPP at 33 months from baseline.

**Results** The total cost of STAMPP was £85 900, equivalent to £818 per school and £15 per pupil. Due to very low uptake of the parental component, we calculated costs of £692 per school and £13 per pupil without this element. Costs per pupil were reduced further to £426 per school and £8 per pupil when it was assumed there were no additional costs of classroom delivery if STAMPP was delivered as part of activities such as personal, social, health and economic education. STAMPP was associated with a significantly greater proportion of pupils experiencing a heavy drinking episode avoided (0.08/8%) and slightly lower public sector costs (mean difference −£17.19). At a notional willingness-to-pay threshold of £15 (reflecting the cost of STAMPP), the probability of STAMPP being cost-effective was 56%. This level of uncertainty reflected the substantial variability in the cost differences between groups.

**Conclusions** STAMPP was relatively low cost and reduced HED. STAMPP was not associated with any clear public sector cost savings, but neither did it increase them or lead to any cost-shifting within the public sector categories. Further research is required to establish if the cost-effectiveness of STAMPP is sustained in the long term.

**Trial registration number** ISRCTN47028486; Results.

> **Strengths and limitations of this study**
>
> ► A broad public sector perspective was adopted for the analysis.
> ► Participant level service use data were obtained directly from the students thus avoiding the need to involve parents/guardians in questionnaire completion.
> ► The study was not specifically powered to detect statistically significant differences in costs or cost-effectiveness.
> ► Only two-thirds of the students had complete cost and outcome data.

## INTRODUCTION

Although the prevalence of alcohol use among children and young people is generally decreasing in the UK, it is high in comparison to many European countries.[1–5] Children and young people who engage in heavy drinking expose themselves to a wide range of negative physical and mental health, social, legal and educational risks.[6–9] As alcohol consumption patterns established during adolescence appear to persist well into adulthood,[10 11] heavy drinking in adolescence may contribute to increased health and social services use, placing a substantial economic burden on society.[12 13] Indeed, considering the external costs imposed on society only, the annual burden of alcohol for England and Wales, Scotland and Northern Ireland (NI) are reportedly £21 billion, £2.1 billion

and £680 million, respectively.[14] Heavy episodic (binge) drinking costs alone have been estimated at £4.86 billion per year (£77 per capita at 2014 prices).[12] Given the complexity and scale of the problem, multiple system-wide interventions are required to address the burden of alcohol-related harm (ARH), including cost-effective prevention interventions to reduce binge drinking in young people.

In addition to environmental prevention and policy-led approaches to alcohol (eg, pricing and marketing restrictions),[15] systematic reviews suggest that some universal alcohol prevention interventions are effective in reducing use in young people.[16] Although existing National Institute for Health and Care Excellence (NICE) guidelines on school-based alcohol prevention (update expected 2019) do not recommend any single prevention programme, they called for partnership working between schools and other stakeholders in efforts to prevent use.[17] NICE suggested that school-based interventions should aim to increase knowledge about alcohol, to explore perceptions about alcohol use and to help develop decision-making skills, self-efficacy and self-esteem. The guideline also highlighted the paucity of evidence from economic evaluations of school-based programmes, although a few additional studies have subsequently emerged (see Hill *et al*[18] for a recent review).

The Steps Towards Alcohol Misuse Prevention Programme (STAMPP) cluster randomised controlled trial (cRCT) was undertaken to assess the effectiveness of a combined school-based universal alcohol harm reduction curriculum and a brief parental intervention, compared with education as normal (EAN)[19–21] in reducing self-reported heavy episodic drinking (HED) and ARHs in adolescents. HED was defined as the consumption of ≥6 units in a single episode for male students and ≥4.5 units for female students. Examples of ARHs include getting into fights, damaging property or poorer school performance. The trial found that the STAMPP intervention was well received by both pupils and school staff and it reduced self-reported HED in the past 30 days at 33-month follow-up with statistically significantly fewer students reporting HED in the intervention group compared with EAN. The intervention did not reduce ARHs associated with own drinking, which were low in both trial arms. There was low uptake of the parental component which comprised a presentation delivered by a trained facilitator at school-based parent evenings, followed up by an information leaflet and survey mailed to all intervention pupils' parents. Only 9% of eligible parents in NI and 2.5% of eligible parents in Scotland attended the evening and only 31% in NI and 18% in Scotland returned the survey. Despite this low uptake, it was uncertain whether the observed intervention effect on HED could be accounted for entirely by the classroom component.

The aim of this paper was to report on the cost-effectiveness of STAMPP compared with EAN in reducing self-reported HED in adolescents.

## METHODS

The STAMPP trial has been described in detail elsewhere.[19 20] It recruited 12 738 participants (intervention=6379, EAN=6359) from 105 schools in NI and the Glasgow and Inverclyde areas of Scotland. All participants were in year 8/S1 (aged 11–12) at baseline (June 2012). We performed an incremental cost-effectiveness analysis (CEA) alongside the trial to estimate the cost per young person experiencing HED avoided due to STAMPP at 33 months. A public sector perspective was adopted for the analysis, which encompassed the costs to local authorities, National Health Service (NHS), Personal Social Services and Criminal Justice Service.

### Outcomes

Consistent with the primary outcome of the study, the primary effectiveness measure was the number of pupils who reported any HED in the previous 30 days at 33 months. This was based on responses to the question *'How often in the past month have you drank 4.5 (female)/6 (male) or more units of alcohol?'* Descriptive statistics were used to summarise the proportion of pupils in each arm reporting a heavy drinking episode.

### Intervention resources use

We calculated the economic cost of STAMPP according to the principle of opportunity cost, that is, we attempted to place a value on the benefits which were foregone by STAMPP being delivered instead of something else. We therefore included the full value of all the resources it used, regardless of whether the resources were directly purchased for the study. The process evaluation carried out alongside the trial established that STAMPP was delivered in most schools as part of their personal, social, health and economic (or local equivalent) provision and that curriculum-based alcohol education activities were found to be minimal in both EAN and intervention schools.[19] We therefore made the very conservative assumption that there were no costs associated with EAN, that is, STAMPP was seen as an additional cost to EAN and the costs of other alcohol education activities were not included in the analysis.

Resources were categorised according to the stage they were used in the research process: planning and preparation for delivery (stage 1), and delivery itself (stage 2), in keeping with other trials of behavioural intervention.[22–24] Pre-start-up resources associated with the development of STAMPP were not included in the analyses as they would not be incurred should the intervention be incorporated into the curriculum in the future. These included the development of the teacher manual and pupil work book content, planning the lessons and the design of materials. Further details are provided in the online supplementary file.

### Students' service use

Data on service use by all pupils from baseline to 33 months were collected using an instrument administered

**Table 1** Unit costs (£) of public sector services

| Service | Unit cost (£) | Source |
|---|---|---|
| **Education** | | |
| School nurse | 50.00 | Unit Costs of Health and Social Care 2014[29] p. 85 |
| School counsellor/guidance teacher | 35.02 | Department of Education Northern Ireland[25] (per 1 hour of teacher time, point 2 of upper pay scale) |
| Intervention teacher | 25.89 | Department of Education Northern Ireland[25] (per 1 hour of teacher time, point 3 of main pay scale) |
| Educational psychologist | 41.00 | Unit Costs of Health and Social Care 2014[29] p. 156 |
| Education welfare officer/home school liaison officer | 27.00 | Unit Costs of Health and Social Care 2014[29] p. 155 |
| **Health** | | |
| GP surgery visit | 46.00 | Unit Costs of Health and Social Care 2014[29] p. 195 |
| GP out of hours | 115.00 | Unit Costs of Health and Social Care 2014[29] p. 191 (home visit unit cost assumed as above) |
| Nurse (other than school nurse) | 13.70 | Unit Costs of Health and Social Care 2014[29] p. 192 (per 15.5 min surgery consultation) |
| Hospital appointment | 109.00 | Unit Costs of Health and Social Care 2014[29] p. 111 |
| Accident and emergency | 233.00 | Unit Costs of Health and Social Care 2014[29] p. 111 (see and treat and convey) |
| Overnight hospital stay | 658.33 | NHS Reference Costs 2013–2014[29] (weighted average length of stay and cost of paediatric non-elective long stays) |
| Psychologist | 50.00 | Unit Costs of Health and Social Care 2014[29] p. |
| Counsellor (other than at school) | 50.00 | Unit Costs of Health and Social Care 2014[29] p. 51 |
| Social worker | 79.00 | Unit Costs of Health and Social Care 2014[29] p. 206 (per 1 hour including travel) |
| Telephone help-line | 3.99 | National Society for the Prevention of Cruelty to Children Annual Report and Account 2014/2015[30] cost per call to Childline deflated to 2013–2014 |
| **Criminal justice** | | |
| Youth justice service | 84.00 | Unit Costs of Health and Social Care 2014[29] p. 224 (face-to-face contact) |
| Police service | 325.00 | Unit Costs of Health and Social Care 2014[29] p. 149 |

GP, general practitioner.

at baseline, 12, 24 and 33 months.[19] The instrument incorporated some items taken from the Client Service Receipt Inventory[25] specifically adapted for childhood[26] and items relating to the use of judicial services. An information page was provided containing definitions of some of the public services in case the students were unfamiliar with them. The instrument was designed with input from relevant professionals (eg, educational psychologist, social workers, Scottish and NI teachers) and reviewed by a social researcher experienced in delivering questionnaires to children and other health economists. The instrument asked pupils to report their use of services in the previous 6 months thus providing service use data for the 6 months prebaseline, 7–12, 19–24 and 28–33 months. If there were any missing fields within the service use questionnaire, it was assumed that the relevant service had not been used.

Individual-level service use was combined with unit costs (table 1) to estimate a cost for each pupil for each of the four survey time periods. Unit costs were obtained from publicly available sources[27–29] and set at 2013–2014. For school counsellors/guidance teacher, we assumed a mid-point of on the Upper Pay Scale (UPS2)[30] and included 24% for employers' costs. As there were gaps in the survey period where no cost data were collected (baseline to 6, 12–18 and 24–27 months), we used interpolation of the available cost data to fill in these gaps. This was done separately for each participant to obtain an estimate of their total service use costs at 33 months. Total costs could not be calculated for those pupils who had any missing cost data points as these were required for the interpolation. Descriptive statistics were used to summarise service use and costs at each time point for the different categories of the public sector (education,

health and criminal justice). Costs of service use in the 6 months before baseline were not included in total costs but were used in the interpolation.

## Cost-effectiveness analysis

Cost-effectiveness was estimated using conventional decision rules[31] and reported as incremental cost-effectiveness ratios (ICERs) if appropriate. The ICER is a measure of the additional cost per additional unit of effect produced by one intervention compared with another. For an intervention to be considered cost-effective, the ICER must be less than the maximum amount of money that a decision-maker would be willing to pay per unit of effect, that is, their willingness-to-pay (WTP) threshold. A CEA estimated the incremental cost per young person experiencing HED avoided due to STAMPP at 33 months. To maintain the correlation structure of the data, only pupils with complete cost and outcome data were included in the analyses. This meant pupils with cost data from baseline to 33 months and a response to the primary outcome variable at 33 months.

Bootstrapped multilevel mixed-effects regression models were used to estimate mean incremental costs and effects (with 95% CI based on 1000 bootstrap resamples). Both models adjusted for school location (NI/Scotland), school level of Free School Meals (FSM) provision (low: 0%–15.4%; moderate: 15.5%–30.4%; high: 30.5% and above), school type (all boys' school/all girls' school/co education school) and clustering. In addition, the cost model adjusted for pupil's baseline costs and the effects model adjusted for pupils' baseline drinking. Uncertainty in the cost-effectiveness measures was investigated by using the 1000 resampled incremental costs and effects to generate 1000 replications of the ICERs. The replicates were plotted on the cost-effectiveness plane and used to construct cost-effectiveness acceptability curves (CEACs). The CEACs showed the probability of STAMPP being cost-effective compared with EAN at different threshold levels of WTP to avoid a pupil experiencing an episode of heavy drinking in the previous 30 days at 33 months. As there is no generally accepted threshold value for cost per pupil experiencing any HED avoided, we looked at range of thresholds, including the cost of the intervention per pupil, and compared our findings with those of other economic evaluations that have been performed in this research area.

All analyses were performed using Stata V.12/IC for Windows. Costs occurring in the second and third year of the study were discounted at 3.5% in keeping with NICE (2013) guidance.[32]

## Sensitivity analyses

A number of one-way sensitivity analyses were performed to test the robustness of the cost-effectiveness findings.
► Missing total costs and outcomes were filled simultaneously using multiple imputation by chained equations. In the multiple imputation model, we included all of the variables that were to be included in the

subsequent multilevel models that is, the treatment variable, baseline costs, baseline drinking, school location, school level of entitlement to free meals and clustering. We also included 6–12 month costs and 6–12 month drinking. We used predictive mean matching for costs and a logit model for the primary outcome. Five imputed data sets were generated and the results combined.
► Total costs were discounted at a rate of 1.5% as suggested by NICE[33] for public health interventions.
► The multilevel models were re-estimated without adjusting for baseline covariates but still adjusting for clustering.
► Since a linear time trend was assumed between data time points, this might have led to total costs and total heavy drinking episodes being underestimated/overestimated if said trend is not appropriate. The impact of increasing and decreasing total costs by 5% were therefore explored.

## RESULTS

A total number of 12 738 pupils took part in the study, with 6379 in the intervention group and 6359 in usual education. There were 11 316 pupils present at the baseline assessment; 5749 in the intervention group and 5567 in EAN. Approximately two-thirds of all pupils had complete cost and outcome data and were included in the analysis. This was similar across groups (intervention=66%; 4189/6379) (EAN=63%; 4037/6359). Of those pupils not included in the analysis (n=4512), 97% had missing cost data, 56% had missing outcome data and 53% were missing both (see online supplementary table S1).

The resources and costs used in the planning and preparation, and delivery of the intervention are presented in the online supplementary table S2. Total costs are presented in table 2. The mean cost per school was £818,

**Table 2** Total cost to deliver STAMPP

| Stage 1: planning and preparation for delivery | Total cost (£) |
|---|---|
| Materials | 6694 |
| Training | 38 079 |
| **Stage 1: subtotal** | **44 773** |
| **Stage 2: delivery** | |
| Teaching | 27 877 |
| Facilitator (for parental component) | 13 250 |
| **Stage 2: subtotal** | **41 127** |
| **STAMPP Total cost** | **85 900** |
| Mean cost/school* | **818** |
| Mean cost/pupil† | **15** |

*Based on 5749 pupils at baseline and 192 classes.
†Based on 105 schools.
STAMPP, Steps Towards Alcohol Misuse Prevention Programme.

**Table 3** Costs (£) at 33 months, by group

| | Intervention, n=4189 | Education as normal, n=4037 | Difference (95% CI) |
|---|---|---|---|
| Costs (£; mean; 95% CI) | | | |
| Education | 287.46 (247.33, 327.58) | 284.81 (244.20, 325.42) | 2.65 (–54.29 to 59.60) |
| Health | 1839.07 (1564.87, 2113.26) | 1906.55 (1627.59, 51) | –67.48 (– 460.16 to 325.19) |
| Criminal | 128.16 (76.67, 179.66) | 101.89 (49.43, 154.35) | 26.27 (–47.55 to 100.10) |
| Total public service costs | 2260.47 (1950.23, 2570.72) | 2292.66 (1977.10, 2608.22) | –32.19 (–476.38 to 412.00) |

Costs discounted at 3.5%. Values are mean (95% CI) adjusted for baseline covariates and clustering.
n, number analysed.

and the mean cost per pupil was £15. The largest proportion of the costs was associated with the training of the teachers as this involved teaching cover, location costs and facilitator costs. The second largest cost was associated with the delivery of the intervention in the classroom setting.

The use of public sector services by all pupils with available data in the 6 months pre-baseline, from 6 to 12 months, from 18 to 24 months and from 27 to 33 months are presented in the online supplementary tables S3-S5. For those pupils with complete data, the costs of public services used over the study period following interpolation of costs and adjustment for baseline covariates and clustering are shown in table 3. Negative costs reflect a cost saving in favour of the intervention. The difference in total public sector costs was small (£) and not statistically significant.

The proportion of pupils reporting a heavy drinking episode in the previous 30 days is shown in table 4. The outcomes are reported in terms of cases avoided; thus, a positive difference reflects a smaller number of pupils in the intervention arm experiencing a heavy drinking episode in the previous 30 days. Statistically significantly fewer pupils in the intervention arm reported drinking heavily in the previous 30 days (15% vs 23%) compared with EAN.

### Cost-effectiveness analysis
The results from the primary CEA are shown in table 5. The cost per pupil for the intervention (£15) was added to each pupil in the intervention arm. STAMPP was associated with a statistically significantly greater proportion of pupils experiencing a heavy drinking episode *avoided* (0.08/8%; 0.06, 0.09) and lower mean total costs (–£17.19;

–402.84, 368.46). The difference in costs was small and not statistically significant; therefore, STAMPP could be considered cost-neutral. When an ICER is negative, as in this case (–17.19/0.08), its magnitude does not convey any meaning so they are not calculated.[34] STAMPP can be said to dominate usual education; however, since the difference in costs was not statistically different, only weak dominance can be claimed.[35] Uncertainty surrounding the estimates of total costs and outcomes is represented by the bootstrapped ICERs on the cost-effectiveness plane (figure 1). The majority of points straddle both the north east and the south east quadrant indicating that, although STAMPP is likely to be more effective than the usual education, there is considerable variability about the cost estimates. The corresponding CEAC is in figure 2; when WTP thresholds ranging from £0 to £800 are considered, it can be seen that the probability of STAMPP being cost-effective compared with usual education ranges from 55% to 67%. Uncertainty in the cost-effectiveness of the intervention remains substantial until much higher WTP values, with an 80% probability being displayed at a WTP of £2000. If decision-makers were only willing to pay the £15 (the cost of STAMPP per pupil), the probability would be 56%.

The results of the sensitivity analyses for the primary CEA are presented in table 5 and the corresponding CEACs are in the online supplementary figure S1. After the multiple imputation of missing data, the probability of STAMPP being cost-effective was lower at each WTP threshold, ranging from 40% to 55%. In light of this, we explored the baseline characteristics of pupils with complete and incomplete data (online supplementary tables S6-S7). Those with missing data were more likely

**Table 4** Proportion of pupils reporting any heavy drinking episode in the previous 30 days at 33 months by group

| Intervention, n=4189 | Education as normal, n=4037 | Difference in effect (proportion of pupils reporting a heavy drinking episode in previous 30 days *avoided*) (95% CI)* |
|---|---|---|
| 0.15 (0.14, 0.17) | 0.23 (0.21, 0.25) | 0.08 (0.06 to 0.09) |

Values are proportion (95% CI) adjusted for baseline covariates and clustering.
n, number analysed.
*CI based on 1000 bootstrap resamples.

**Table 5** Results of the primary cost-effectiveness analysis and sensitivity analyses

| Analysis | Intervention, n (% total) | Education as normal, n (% total) | Difference in total costs (£) (intervention+total public service costs) | Difference in effect (proportion of pupils reporting a heavy drinking episode in previous 30 days *avoided*) |
|---|---|---|---|---|
| Primary analysis at 33 months | 4189 (65.7) | 4037 (63.5) | −17.19 (−402.84, 368.46) | 0.08 (0.06, 0.09) |
| Sensitivity analysis for primary cost-effectiveness analysis at 33 months | | | | |
| Multiple imputation for missing cost and outcome data | 6379 (100) | 6359 (100) | 34.10 (−299.44, 367.44) | 0.08 (0.07, 0.09) |
| Discounting costs and outcome at 1.5% | 4189 (65.7) | 4037 (63.5) | −17.63 (−410.62, 375.36) | 0.07 (0.06, 0.09) |
| Adjustment of costs and outcome data for cluster only | 4189 (65.7) | 4037 (63.5) | −49.23 (−419.52, 321.07) | 0.08 (0.06, 0.10) |
| 5% increase in costs | 4189 (65.7) | 4037 (63.5) | −18.05 (−422.98, 386.88) | 0.07 (0.06, 0.09) |
| 5% decrease in costs | 4189 (65.7) | 4037 (63.5) | −16.33 (−382.70, 350.04) | 0.07 (0.06, 0.09) |

Total participants in the study=12 738 (intervention=6379, education as normal=6359).
n, number analysed .

to go to a school with a high proportion of free school meals, more likely to have reported HED in the previous 30 days at baseline and reported marginally higher public service use in the 6 months prebaseline, compared with those pupils with complete data.

When the multilevel models were re-estimated adjusting only for baseline covariates, the probability of cost-effectiveness was consistently higher, ranging from 65% to 76%. Reducing the discount rate to 1.5% and increasing/decreasing costs by 5% had little effect on the cost-effectiveness of STAMPP.

## DISCUSSION
The total cost to deliver STAMPP was £85 900, equivalent to £818 per school and £15 per pupil. In a review of economic evidence for the development of

NICE guidance,[17] the cost of other effective school-based interventions range from £20 to £150 (cost year 2005/2006),[36–38] one of which was Australian School Health and Alcohol Harm Reduction Project (SHAHRP) from which the classroom component of STAMPP was adapted.[36] Another review of school-based skills development substance use prevention curricula in the USA (which targeted both alcohol and illicit drug estimated programme unit costs between US$100 and US$400 per pupil[39]). Thus, at a cost of £15 per pupil, STAMPP is a relatively low-cost intervention that successfully

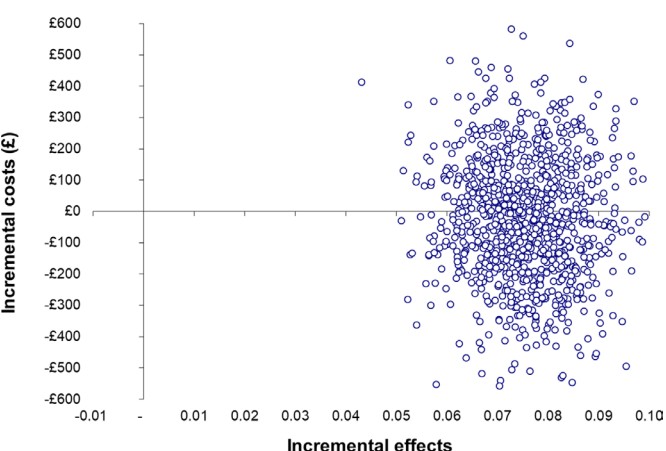

**Figure 1** Cost-effectiveness plane for the primary cost-effectiveness analysis showing bootstrapped replications of mean incremental costs and pupils experiencing heavy drinking avoided.

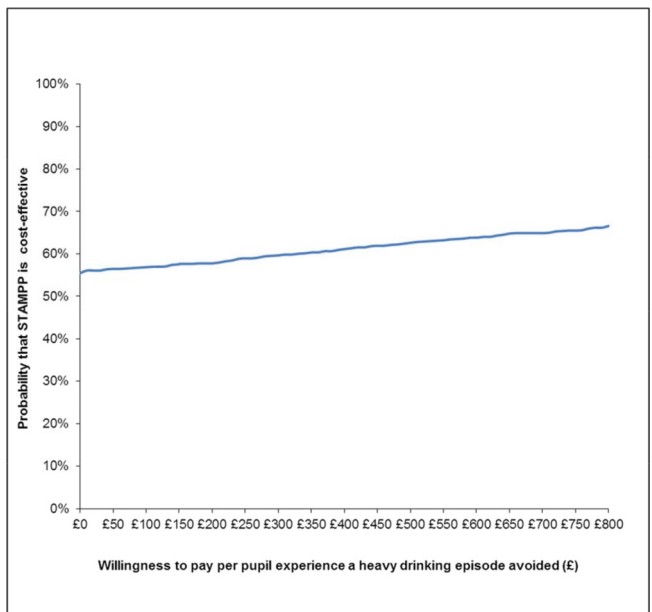

**Figure 2** Cost-effectiveness acceptability curve showing the probability of Steps Towards Alcohol Misuse Prevention Programme (STAMPP) being cost-effective compared with education as normal.

reduces problematic drinking. If costs were extrapolated to the 685 300 students aged 12 years old in the UK in 2013/2014, the cost would be £10.3 million in total. This is a relatively small amount compared with the overall annual economic burden from alcohol consumption in the UK. Furthermore, the cost per pupil could reduce to £8 per pupil if it was entirely classroom based and there was no additional cost for teachers' time to deliver it (see online supplementary table S8). This is justifiable considering the low uptake of the parental component and since we made the conservative assumption that STAMPP was delivered in addition to EAN; in reality, STAMPP is likely to have replaced whatever statutory provision was in place, however minimal, and potentially saved teachers' time as observed in a recent cRCT of a school-based sexual education intervention.[24]

Costs were estimated from a public sector perspective, which was justified considering one of the principal objectives of STAMPP was to reduce ARHs in teenagers. It was hypothesised that this would in turn reduce the use of health and judicial services and the need for additional support within the school setting to address behavioural and emotional problems. The analysis of public service costs however showed only a small difference between groups at 33 months. This is in keeping with the analysis of the ARH data from the trial as no difference was observed in the number of self-reported ARHs by pupils between groups, and indeed, both groups reported low levels of harms overall.[19 20]

Compared with EAN, significantly fewer pupils in the intervention group reported experiencing a heavy drinking episode in the previous 30 days at 33 months. The CEA indicated that STAMPP weakly dominated EAN. At a notional WTP threshold of £15 (reflecting the cost of STAMPP), the probability of STAMPP being cost-effective was 56%. This level of uncertainty reflects the considerable variability in the cost differences between groups.

The sensitivity analyses indicated that the results of the CEA were robust to small changes in parameters, that is, discounting and small increases in cost and effectiveness. However, when costs and effects were not adjusted for baseline covariates, the probability of cost-effectiveness of STAMPP increased. This suggests that the cost-effectiveness of STAMPP may vary between subgroups and warrants further investigation to identify which pupils or schools might benefit the most from receiving the intervention. Furthermore, when multiple imputation was used to impute missing cost and outcome data, the cost-effectiveness of STAMPP decreased. Multiple imputation is based on the assumption that data are missing at random; however, a post hoc comparison of the baseline characteristics of pupils with complete and incomplete data indicated the assumption might not have been met. Pupils with incomplete data were more likely to attend a school with a high proportion of free school meals, more likely to report HED at baseline and reported higher public service use at baseline, compared with those with complete data.

The strengths of the economic evaluation include the choice of perspective; we adopted a public sector perspective in keeping with NICE guidance for public health interventions,[33] thus we looked beyond the healthcare system and considered the broader impact of the intervention. In contrast, Hill *et al*[18] found that the majority of economic evaluations of alcohol prevention considered only healthcare costs; the danger being that interventions may be undervalued if cost savings occur in other sectors or overvalued if costs are incurred in other sectors. A further strength was that we obtained resource use data directly from the pupils, thus avoiding the need to involve parents/guardians in questionnaire completion. We provided definitions of the services using age appropriate terminology, with input from relevant professionals. Considering the difficulty associated with engaging parents/guardians in the study, reflected in the poor attendance to the parental evenings,[19] it is likely that a reliance on parents/guardians would have led to considerable amounts of missing data.

The study had a number of limitations. We relied on self-report to collect our outcomes, which may have led to under-reporting or over-reporting of alcohol use through memory, social desirability and other biases.[40] However, adolescent self-reported alcohol questionnaires are generally reliable[41] and a low level (9.9%) of recanting (the denial of a previous positive report of lifetime[42]) was observed in this study.[43] Furthermore, all pupils completed the same questionnaires, so if bias had existed, this would arguably have been equivalent between trial arms. In terms of self-report service use, cost data were more likely to be missing than outcome data. This likely reflects the need for pupils to complete multiple questionnaires over the study period in order to calculate total costs, whereas only a response to a single question at the final time point was required for outcome data.

The study was not specifically powered to detect statistically significant differences in costs or cost-effectiveness. Although CEA does not typically make decisions based on significance rules,[44] having a sufficiently powered study will allow decision-makers to be more confident in the value claim.[34] The resources used during the planning, preparation and delivery of the intervention were largely recorded retrospectively and costs were obtained from invoices when these were available. We endeavoured to use plausible assumptions when actual data were not available, but the consistent and prospective collection of resource use and costs would lead to more robust data. We included pupils in the CEA only if they had complete cost and effect data. As a result, only two-thirds of the pupils were included in the analysis, the rest we assumed were missing at random. As discussed earlier, this assumption may be flawed. We performed a within trial CEA which is limited to the time horizon of the study (33 months). In light of the literature linking excess and early initiation of drinking in adolescence to alcohol use behaviours in adulthood (eg, the development of poor health outcomes over a sustained period of time,[45] it is important to investigate

if the (cost) effectiveness of STAMPP is maintained or even increases in the long term. A decision model was beyond the scope of this current study; however, it would provide a framework to explore the long-term impact of STAMPP. Finally, we did not undertake a formal analysis of the impact of STAMPP on health inequalities, but an equity impact analysis[46] that disaggregates costs and outcomes by equity-relevant subgroups such as gender, receipt of free school meals, school area deprivation and ethnicity may help us to understand this impact better.

## CONCLUSIONS

STAMPP was a relatively low-cost intervention that successfully reduced HED. STAMPP did not bring about clear public sector cost savings; however, neither did it increase them or lead to any cost-shifting within the public sector categories. STAMPP can therefore be considered to weakly dominate EAN because it was both cost-neutral and more effective. Further research is required to establish if the cost-effectiveness of STAMPP is sustained in the long term.

## Patient and public involvement

Pupils participating in this trial were not involved in its design. The research questions and outcome measures used in this study were not directly informed by pupils' priorities, experience and preferences. They were partly aligned to the funding call of the grant awarding body. However, this trial followed an earlier pilot study in NI that adapted and evaluated the Australian SHAHRP for delivery in the UK, and was informed by pupil and teacher experiences of that research. Education professionals including school head teachers, principals and subject leads facilitated the participation of pupils in the study, but pupils were not involved in the recruitment and delivery of the study. Study results have been disseminated at a number of special events in the study sites. Experiences of participation in the trial formed part of our process evaluation and this was published as part of our report to funders.[19]

**Author affiliations**
[1]Northern Ireland Clinical Trials Unit, The Royal Hospitals, Belfast, UK
[2]Department of Psychological Sciences, University of Liverpool School of Life Sciences, Liverpool, UK
[3]Psychology and Public Health, Oxford Brookes University, Oxford, UK
[4]School of Sport, Health and Exercise Sciences, University of Bangor, Bangor, UK
[5]School of Social Sciences, Education and Social Work, Queen's University Belfast, Belfast, UK
[6]Public Health Institute, Liverpool John Moores University, Liverpool, UK

**Acknowledgements** As well as acknowledging the role played by participating schools and school children, the authors would like to acknowledge the support of the following people in this project: Séamus Mullin, Gerry Bleakney, Owen O'Neill (Public Health Agency of Northern Ireland); Malachy Crudden (CCMS), Maura Kearney, and Fergal Doherty (Psychological Services, Glasgow); Kate Watson (Psychological Services, Inverclyde) and John Butcher and Sandy Cunningham (Education Services, Glasgow). Thank you also to Rita Faria (Centre for Health Economics, University of York) for her valuable feedback on an early draft of the manuscript. The views and opinions expressed therein are those of the authors and do not necessarily reflect those of the National Institute of Health Research (NIHR)-Public Health Research, NIHR, National Health Service or the Department of Health.

**Contributors** HS had full access to all of the data in the study and takes responsibility for the integrity of the data and the accuracy of the data analysis. AA undertook the health economic analysis, wrote the first draft of the manuscript and subsequent versions, and submitted the final version; HS was project principal investigator, contributed to the first draft and subsequent iterations of the manuscript; JC, PD, DF, SH, MMcK, LM and AP all contributed to drafts and approved the submission.

**Funding** This trial was funded by the National Institute of Health Research Public Health Research programme (project number 10/3002/09). The Public Health Agency of Northern Ireland and Education Boards of Glasgow/Inverclyde provided some intervention costs. Diageo provided funds to print classroom workbooks for use only in the Glasgow Local Authority area. Remaining intervention costs were internally funded.

**Disclaimer** The views and opinions expressed therein are those of the authors and do not necessarily reflect those of the National Institute of Health Research (NIHR)-Public Health Research, NIHR, National Health Service or the Department of Health. The research and intervention funders had no involvement in intervention design; design and conduct of the study; collection, management, analysis and interpretation of the data; and preparation, review or approval of the manuscript.

**Competing interests** The sponsor University (LJMU) received and administered a payment from the alcohol industry for printing of student workbooks in the Glasgow trial site only. Percy reported that he has previously received funding from the European Foundation of Alcohol Research (ERAB) in relation to the development of statistical models for longitudinal data (2008-2010). Foxcroft reported that his department has previously received funding from the alcohol industry for unrelated prevention programme training work. Sumnall reported that his department has previously received funding from the alcohol industry (indirectly via the industry funded Drinkaware charity) for unrelated primary research.

**Patient consent for publication** Obtained.

**Ethics approval** The research was approved by Liverpool John Moores University Research Ethics Committee (11/HEA/097). Consent was obtained from school head teachers/principals before randomisation. Consent from participants and their parents/guardians was obtained after randomisation.

**Provenance and peer review** Not commissioned; externally peer reviewed.

**Data sharing statement** The datasets generated during and/or analysed during the current study are not yet publicly available due to the authors undertaking additional analyses and follow-on studies, but are available from the corresponding author on reasonable request.

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
