## [Reviewer comments · BMJ Open]

ARTICLE DETAILS

TITLE (PROVISIONAL)	Cost-effectiveness of a combined classroom curriculum and parental intervention: economic evaluation of data from the Steps Towards Alcohol Misuse Prevention Programme (STAMPP) cluster randomised controlled trial.
AUTHORS	Agus, Ashley; McKay, Michael; Cole, Jonathan; Doherty, Paul; Foxcroft, David; Harvey, Séamus; Murphy, Lynn; Percy, Andrew; Sumnall, Harry

VERSION 1 - REVIEW

REVIEWER	Professor Simpn Coulton University of Kent UK Previous funding from NIHR PHR and NIHR PGfAR to explore school based and ED based interventions to reduce alcohol consumption in adolescent populations.
REVIEW RETURNED	20-Dec-2018

GENERAL COMMENTS	The paper reports on the economic analysis of a large cluster randomised controlled trial of school-based curriculum-based interventions to address alcohol misuse in an adolescent population. Overall the paper is well written, aims and objectives and methods are clear and appropriate and the conclusions justified by the analysis. I had a few minor points that the authors may wish to address 1. A sentence could be added in paragraph 3 of the introduction informing readers of the primary findings of the STAMPP trial to allow readers who may not be familiar with the study to understand the context.2. At the end of paragraph 3 of the introduction the statement "...it was concluded that effects on harms might manifest later." This statement is not substantiated by the research and emerging evidence in this population suggests that alcohol-related harm is mediated by other factors, such as conduct disorder and emotional dysregulation, rather than alcohol consumption alone. It might be worth considering deleting the phrase.3. In outcomes a reference for the definition of HED might be useful, the definition heavy episodic drinking in this population often differs across studies.4. There are some instances where references to tables have links missing but that is clearly a minor formatting error.
--

REVIEWER	David Kim Tufts Medical Center
REVIEW RETURNED	13-Feb-2019

GENERAL COMMENTS	General comments: Thank you for the opportunity to review this paper. Authors conducted a trial-based cost-effectiveness analysis of STAMPP (alcohol misuse prevention program among school-aged students), compared to the status quo (education as normal). Although the short evaluation period (33 months) may not long enough to capture all of important consequences of reducing heavy drinking among adolescents, the study finding could be useful to inform policy discussion on preventing heavy drinking among adolescents. Also, looking at the impact on multiple public sectors was helpful. Major points: As one-third of the participants reported missing either costs or outcome data (which is not able to differentiate based on the data provided), it seems very important to investigate the missing data patterns. Also, your sensitivity analyses after multiple imputation shows substantial reductions in probability of being cost-effective for the STAMPP by 15 percentage points. So, I would suggest authors to compare baseline characteristics of those with missing information with those without missing information to test “missing at random” assumptions. Also, the characteristics of those with missing data should be compared between the intervention and the control (EAN) groups. Finally, I would make the imputed data analysis as a base case while treat the complete case analysis as a sensitivity analysis unless the authors have a compelling reason to argue that missing data are not a problem. It was confusing in the way the authors described alternative costs of delivering the STAMPP (line 268-273) in the method section (which is not a part of main analysis). What does it mean that “in the light of low uptake~, we also present the costs of delivering the STAMPP classroom component only?” Because this is an ex-post analysis of the clinical trial, what actually provided in the trial should be used in the base case analysis. Does the original trial have the separate classroom component only and materials and teacher training only? (if so, please note explicitly) Or is this something the authors tried to provide implications for future works because the original component in the trial was not well received (i.e., low uptake among parents). If it is the latter case, the other scenarios should be exploratory, rather than a trial-based, and explicitly mentioned in the discussion section, rather than the main result section, as suggesting alternative approaches to improve cost-effectiveness. Main results (Table 3) seem puzzling to me and need better explanation of what drives the results. The overall cost savings are primarily driven by reductions in health care costs among the intervention group (-\$67), but the intervention group spent more on education (\$2.65) and criminal justice system (\$26) on average. Also, this is a bit contradictory with what authors interested in the discussion section. (line 342-346) – Here, why suddenly focus on the statistically insignificance in the interpreting this results, rather
---

	than the average, as the decision-making generally based on the expected outcomes, not a statistical significance. Finally, more discussion is needed on the limitations of trial-based CEA, mainly due to short-term time horizon. Particularly, the impact of reduced heavy drinking on health care and criminal justice system may not be captured appropriately in the short evaluation period (33 months) Also, the potential bias and implications of using “self-reported outcomes” should be discussed. (e.g., potentials for underreporting?) Minor points: I suggest moving details on costing (line 139-line 174) to online appendix and summarize the primary data and approach in one paragraph. Table 5. The fourth column should be labeled as Total Costs (Intervention Costs + Total Public Service Costs), and please provide a brief description of the negative costs. (i.e., costs saved) Formatting issues: Need to correct the issues with references (different styles are used (sometimes appeared twice), e.g., line 68-69, 72-23, 278) Line 67: exposure -> expose? Line 73: double periods. Line 82: “environmental and policy” does not make sense Line 86: expected update in 2019? Line 96: move “compared with education as normal (EAN)” from line 99 Line 98: Please define heavy episodic drinking - defined in line 119-120, but would be great to bring this up when HED first appears. Line 100: Please describe briefly on what are the specific parental components, why compliance was low, and what is the impact of low parent compliance on the outcome Line 101: Give specific examples of alcohol-related harms. Line 118: remove economic Line 142-143: unclear on what it means
--	---

VERSION 1 – AUTHOR RESPONSE

Reviewer(s)' Comments to Author:

Reviewer: 1

Reviewer Name: Professor Simon Coulton

Institution and Country: University of Kent UK Please state any competing interests or state 'None declared': Previous funding from NIHR PHR and NIHR PGfAR to explore school based and ED based interventions to reduce alcohol consumption in adolescent populations.

Please leave your comments for the authors below The paper reports on the economic analysis of a large cluster randomised controlled trial of school-based curriculum-based interventions to address alcohol misuse in an adolescent population.

Overall the paper is well written, aims and objectives and methods are clear and appropriate and the conclusions justified by the analysis.

@@Thank you.

I had a few minor points that the authors may wish to address

1. A sentence could be added in paragraph 3 of the introduction informing readers of the primary findings of the STMPP trial to allow readers who may not be familiar with the study to understand the context.

@@ Further details of the primary findings have now been added to this paragraph. We have also provided some detail on the parental component as requested by the other reviewer.

2. At the end of paragraph 3 of the introduction the statement "...it was concluded that effects on harms might manifest later." This statement is not substantiated by the research and emerging evidence in this population suggests that alcohol-related harm is mediated by other factors, such as conduct disorder and emotional dysregulation, rather than alcohol consumption alone. It might be worth considering deleting the phrase.

@@This has now been deleted.

3. In outcomes a reference for the definition of HED might be useful, the definition heavy episodic drinking in this population often differs across studies.

@@Very good point thanks you- we have actually incorporated this into the 3rd paragraph of the introduction so the information is there from the outset.

4. There are some instances where references to tables have links missing but that is clearly a minor formatting error.

@@ We have corrected these errors now thank you.

Reviewer: 2

Reviewer Name: David Kim

Institution and Country: Tufts Medical Center Please state any competing interests or state 'None declared': None

Please leave your comments for the authors below General comments:

Thank you for the opportunity to review this paper. Authors conducted a trial-based cost-effectiveness analysis of STAMPP (alcohol misuse prevention program among school-aged students), compared to

the status quo (education as normal). Although the short evaluation period (33 months) may not long enough to capture all of important consequences of reducing heavy drinking among adolescents, the study finding could be useful to inform policy discussion on preventing heavy drinking among adolescents. Also, looking at the impact on multiple public sectors was helpful.

Major points:

As one-third of the participants reported missing either costs or outcome data (which is not able to differentiate based on the data provided), it seems very important to investigate the missing data patterns. Also, your sensitivity analyses after multiple imputation shows substantial reductions in probability of being cost-effective for the STAMPP by 15 percentage points. So, I would suggest authors to compare baseline characteristics of those with missing information with those without missing information to test “missing at random” assumptions. Also, the characteristics of those with missing data should be compared between the intervention and the control (EAN) groups. Finally, I would make the imputed data analysis as a base case while treat the complete case analysis as a sensitivity analysis unless the authors have a compelling reason to argue that missing data are not a problem.

@@ Thank you for highlighting this. In terms of differentiating between missing cost and outcome data we now report on the percentage of pupils with missing cost, outcome and cost & outcome data on the first paragraph of the results section and in Table S1 of the supplementary file. We comment on this in the discussion (Lines 344-347). We also now include two tables comparing the baseline characteristics of pupils with complete and incomplete cost/outcome data, by trial arm in the supplementary file (Tables S7 & S8) and report some observed differences in Lines 289-293 of the results. We have not made the imputed data analysis the base-case analysis as you suggest because this would deviate from our Data Analysis Plan (DAP). We stipulated that multiple imputation of missing data would be performed as a sensitivity analysis. The DAP was reviewed and approved by the trial steering committee and signed-off by the chief investigator and trial statistician prior to analysis commencing on the study.

It was confusing in the way the authors described alternative costs of delivering the STAMPP (line 268-273) in the method section (which is not a part of main analysis). What does it mean that “in the light of low uptake~, we also present the costs of delivering the STAMPP classroom component only?” Because this is an ex-post analysis of the clinical trial, what actually provided in the trial should be used in the base case analysis. Does the original trial have the separate classroom component only and materials and teacher training only? (if so, please note explicitly) Or is this something the authors tried to provide implications for future works because the original component in the trial was not well received (i.e., low uptake among parents). If it is the latter case, the other scenarios should be exploratory, rather than a trial-based, and explicitly mentioned in the discussion section, rather than the main result section, as suggesting alternative approaches to improve cost-effectiveness.

@@ We have now provided more information for the reader on the parental component of the study and its low uptake by parents in the 3rd paragraph of the introduction (Lines 100-106). We did present alternative intervention costs to provide additional information for stakeholders post hoc, so have moved reference to this to the discussion as you suggest (Lines 312-318)

Main results (Table 3) seem puzzling to me and need better explanation of what drives the results. The overall cost savings are primarily driven by reductions in health care costs among the intervention group (-\$67), but the intervention group spent more on education (\$2.65) and criminal justice system (\$26) on average. Also, this is a bit contradictory with what authors interested in the discussion section. (line 342-346) – Here, why suddenly focus on the statistical insignificance in the interpreting this results, rather than the average, as the decision-making generally based on the expected outcomes, not a statistical significance.

@@ Many thanks for highlighting our inconsistencies. We have now heavily edited the relevant discussion paragraph (Lines 320-327). We have opted not to explore the costs differences any more than we have- the differences are small with wide ranging 95% confidence intervals. The scatterplot on the cost-effectiveness plane highlight the uncertainty surrounding the incremental costs quite clearly.

Finally, more discussion is needed on the limitations of trial-based CEA, mainly due to short-term time horizon. Particularly, the impact of reduced heavy drinking on health care and criminal justice system may not be captured appropriately in the short evaluation period (33 months) Also, the potential bias and implications of using “self-reported outcomes” should be discussed. (e.g., potentials for underreporting?)

@@ These are very important limitations to raise thank you. Our discussion has now been edited to highlight both of these issues.

Minor points:

I suggest moving details on costing (line 139-line 174) to online appendix and summarize the primary data and approach in one paragraph.

@@ We have made the change you suggest, the information is not in the online supplementary file.

Table 5. The fourth column should be labeled as Total Costs (Intervention Costs + Total Public Service Costs), and please provide a brief description of the negative costs. (i.e., costs saved)

@@ Corrected now, thank you.

Formatting issues:

Need to correct the issues with references (different styles are used (sometimes appeared twice), e.g., line 68-69, 72-23, 278)

@@ These have now been corrected.

Line 67: exposure -> expose?

Line 73: double periods.

Line 82: "environmental and policy" does not make sense Line 86: expected update in 2019?

Line 96: move "compared with education as normal (EAN)" from line 99 Line 98: Please define heavy episodic drinking - defined in line 119-120, but would be great to bring this up when HED first appears.

@@ Thank you, we have now defined this in the introduction (Lines 93-95).

Line 100: Please describe briefly on what are the specific parental components, why compliance was low, and what is the impact of low parent compliance on the outcome

@@ Details have now been provided in the introduction (Lines 100-106).

Line 101: Give specific examples of alcohol-related harms.

@@ These have been included now (Lines 95-96)

Line 118: remove economic

@@ Done

Line 142-143: unclear on what it means

@@ Adopted in to the curriculum. This has been clarified.

VERSION 2 – REVIEW

REVIEWER	Simon Coulton University of Kent UK
REVIEW RETURNED	05-May-2019

GENERAL COMMENTS	The authors have satisfactorily addressed the comments made on the previous review
--